# Effects of dietary of *Bacillus coagulans*, whey powder, and their interaction on the performance of Lohmann LSL-lite laying hens in the late production phase

Zahra Hamzehee[1], Mehran Torki[1]*, Khodabakhsh Rashidi[2], Alireza Abdolmohammadi[1]

**1** Department of Animal Science, College of Agriculture and Natural Resources, Razi University, Kermanshah, Iran, **2** Research Centre of Oils and Fats, Kermanshah University of Medical Sciences, Kermanshah, Iran

* torki@razi.ac.ir

## Abstract

Due to the need to produce high-quality and healthy eggs, the current experiment was conducted to investigate the impact of dietary supplementation of whey powder (WP), *Bacillus coagulans (B. coagulans)*, and their combination (MIX) on the production performance, egg quality, blood biochemical parameters, and histomorphological parameters of Lohmann LSL-lite laying hens. 144 Lohmann laying hen (75 weeks) were randomly assigned to 4 different dietary treatments, with 6 replications and 6 hens per cage. The hens were fed a basal diet (control, CON), the basal diet supplemented with 1 g/kg WP, 1 g/kg *B. coagulans* ($4 \times 10^6$ CFU), and 1 g/kg WP plus 1 g/kg *B. coagulans* probiotic for 12 weeks. Feed intake and egg weight were not affected by the treatments at any stage of the trial ($P > 0.05$). No significant interaction was found between WP and *B. coagulans* in egg quality parameters and blood-biochemical parameters other than malondialdehyde ($P < 0.05$). The level of malondialdehyde in serum was reduced when WP was used along with *B. coagulans* compared to when WP was used alone. However, egg production in all periods and egg mass in the first period were affected by the synergistic effect of WP and *B. coagulans*. Furthermore, FCR was reduced in the first period (75–80 weeks) under the influence of the MIX group compared to the control group or when used alone ($P < 0.05$). The color of the yolk was increased in the group receiving *B. coagulans* compared to the control group ($P < 0.05$). Therefore, in birds fed with *B. coagulans*, a significant increase in the width of the villi in the ileum was observed ($P \leq 0.05$). Interestingly, *B. coagulans* and WP reduced performance when used alone compared to the control, but improved performance when combined. In conclusion, the simultaneous use of WP and *B. coagulans* in diet can probably improve the parameters of production performance, FCR, and serum malondialdehyde level at the end of the production period of Lohmann laying hens.

**Data availability statement:** All relevant data are within the manuscript.

**Funding:** The author(s) received no specific funding for this work.

**Competing interests:** The authors have declared that no competing interests exist.

## Introduction

Antibiotic resistance (ABR) has increased over the past century due to the overuse of antibiotics for therapeutic purposes and as growth promoters in poultry production [1]. ABR is a global problem that is not limited to any country, and no country can stop its progress alone [2]. The United States, Brazil, the European Union, and China remain the largest poultry meat producers in the world [3]. In order to combat ABR and other negative effects caused by the indiscriminate use of antibiotics, the European Union banned the use of antibiotics in animal feed in 2006 [4]. The US Food and Drug Administration issued a rule in 2015 to restrict the judicious use of antibiotics only in the treatment of animals under veterinary feed guidelines [5]. Hence, increased attention has been drawn to the research and development of antibiotic substitutes such as prebiotics, probiotics and postbiotics [6–10], botanical compounds [11,12], waste materials such as by-products, oilseeds [12], and exogenous enzymes [13,14].

Furthermore, the laying period from 45 to 72 weeks of age is known as the late phase of hen laying [15]. Aging has negative effects on egg production of laying hens after the peak laying stage [16]. As the age increases, the function of the digestive organs such as the pancreas, liver, and intestine decreases imperceptibly, which results in impaired nutrient ingestion and absorption [17]. Also, the quality of the eggshell in older chickens decreases [18]. In addition, in laying hens, the quality of the egg decreases with age, especially during the late laying cycle [19–21]. Older chickens are more vulnerable to immune challenges [22], and their liver antioxidant capacity is also reduced compared to younger chickens [23]. Eggshell quality parameters, including thickness and breaking strength, gradually decrease during the production cycle [24]. Reduced intestinal barrier function and compromised digestive physiology in aging laying hens could be one of the main reasons for poor egg quality in older hens [25]. A study by Gu et al. [26] showed that eggs harvested from 75-weeks-old hens had lower eggshell strength and thickness, indicating the frangibility of eggshells from older laying hens. *B. coagulans* is a type of gram-positive, spore-forming bacillus that produces lactic acid and does not contain enterotoxin [27]. This *bacillus* is also protected by a protein coating similar to spores, which enables it to withstand high temperatures. In the experiment conducted Acuff and Aldrich [28] on the resistance of *B. coagulase* spores, it was observed that the viability of the spores through three dryer conditions (49 °C for 10 min; 107 °C for 16 min; and 66 °C for 46 min) was similar. Also, it can survive stomach acid and bile salts. This allows it to reach the small intestine, where it can germinate and multiply [27,29,30]. As a result, *B. coagulans* has been identified as a potential probiotic and has been utilized in cattle, broilers, and weaned piglet production [31–33]. According to a study conducted by Zhen et al. [34], the inclusion of *B. coagulans* in broiler diets resulted in improved growth performance and intestinal microbiota as well as a better villus structure of broilers. Additionally, it helped to repair intestinal inflammation and structural damage caused by *Salmonella enteritis* infection. Riazi et al. [35] reported that lactosporin, an antimicrobial substance produced by *B. coagulans*, inhibits some pathogenic microorganisms. A study conducted by Xu et al. [36] has shown that the dietary supplementation

of *B. coagulans X26* to laying hens resulted in a significant improvement in egg quality and production performance. The improvements were attributed to an increase in the villus height of the ileum and the ratio of villus height to crypt depth, as well as alterations in the composition of the intestinal microbiota and the content of short-chain fatty acids (SCFAs) in the intestinal contents.

Whey powder (WP) is a by-product of the dairy industry that is rich in high-quality protein and contains significant amounts of lactose, almost 70% on a dry matter basis [37,38]. Lactose is not digestible in the small intestine of poultry. Instead, it passes down to the lower bowel where beneficial bacteria ferment it and produce SCFAs like butyrate, propionate, and acetate [38,39]. Mehri et al. [40] reported improved FCR by adding 4% WP to the broiler diet. Gülşen et al. [41] illustrated that adding 3.85% WP to broiler feed increased their body weight, although it did not affect feed intake and FCR. A study conducted by Caugant et al. [42] on predominant calves showed that WP seemed to be satisfactorily digested in the small intestine of the predominant calf. Also, in an experiment conducted by Hanson et al. [43] on salmon, they concluded that the benefits of the dietary inclusion level of 5.1 g/kg Whey Protein hydrolysate under normal conditions include 9% higher weight gain and feed consumption.

In this study, according to the prebiotic effects of lactose present in WP, it was assumed that the probiotic *B. coagulans* added to the diet increases the beneficial effects of WP synergistically. So, this study aims to investigate *B. coagulans* and whey powder's main effects and their interaction in the diet, to clarify their effects on production performance, egg quality, blood biochemical parameters, and ileum histomorphology in laying hens at the late production phase, which is of great importance for promoting the healthy breeding of laying hens and the production of safe and high-quality eggs, especially during the late laying phase.

## Materials and methods

### Birds and diets

The research took place at the faculty of agriculture at Razi University in Kermanshah, Iran. All experimental procedures were approved by the Animal Welfare Committee of Razi University (IR.RAZI.REC.1402.070). The total number of 300 fertile eggs were bought from a local hatchery company and set in the laboratory incubator of Animal Science Department. The hatched chicks were then reared until the starting the experimental trail based on the company catalogue. Finally, the total number 144 laying hens were randomly selected from this larger herd and distributed between 24 cages (n = 6 per each replicate cage). So, the used laying hens were not privately owned by another institution/individual/farm. A completely randomized design was used in this experiment. A total of 144 Lohmann layers hens (75 weeks old) were randomly assigned to 4 treatments with 6 replicates of 6 hens in each cage, for 12 weeks. The research examined the effects of *B. coagulans* $4 \times 10^6$ CFU/g (purchased from the Parsilact company) and WP at a rate of one gram per kilogram. The birds were fed a basal diet (control, CON), the basal diet supplemented with 1 g/kg WP, 1 g/kg *B. coagulans*, and 1 g/kg WP further 1 g/kg *B. coagulans* ($4 \times 10^6$ CFU) probiotic. The basal diets are shown in Table 1.

### Management and production parameters

Throughout the experiment, the hens were provided with feed and water ad libitum, and the house temperature was between 20 and 22°C. The light schedules were similar to the guidelines set in the Lohmann Commercial Management Guide. The hens were individually weighed at the start and end of the experiment. Using these values, BW gain was calculated. Feed intake was recorded at 7-d intervals. Feed intake was recorded at the beginning and the end of the experiment and calculated as grams per hen per day. The FCR was calculated as kilograms of feed per kilogram of egg (total feed intake/total egg mass). All the eggs were gathered and weighed individually to determine their weight. The values were calculated based on egg production, egg weight, and egg mass.

**Table 1. Composition and nutrient contents of basal diet fed in the experimental diet of laying hens.**

| Ingredients (%) | | Calculated amount | |
|---|---|---|---|
| Corn | 58.33 | ME (Kcal/kg) | 2710 |
| Barley | 2.00 | CP (%) | 15.48 |
| Soybean meal | 24.00 | CF (%) | 2.90 |
| Rice bran | 1.83 | EE (%) | 8.45 |
| Nacl | 0.22 | Lys (%) | 0.70 |
| Vit & min Premix | 0.60 | Met (%) | 0.40 |
| DL-Methionine | 0.20 | Met + Cys (%) | 0.64 |
| bicarbonate sodium | 0.18 | Thr (%) | 0.50 |
| DCP | 1.23 | AP (%) | 0.31 |
| Oyster | 6.00 | Sodium | 0.15 |
| Calcium carbonate | 3.63 | Calcium | 4.02 |
| Sunflower Oil | 1.66 | | |
| Acidifier | 0.01 | | |
| Phytase | 0.03 | | |
| toxin Binder | 0.08 | | |

Vitamin mixture per 0.3 kg/100 kg of diet: Vitamin A, 7,700,000 IU; Vitamin D3, 3,300,000 IU; Vitamin E, 6600 mg; VitaminK3, 550 mg; thiamine, 2200 mg; riboflavin, 4400 mg; Vitamin B6, 4400 mg; capantothenate, 550 mg; nicotinic acid, 200 mg; folic acid, 110 mg; choline chloride, 275,000 mg; biotin, 55 mg; Vitamin B12, 8.8 mg. [b]Mineral mixture per 0.3 kg/100 kg of diet: Mn, 66,000 mg; Zn, 66,000 mg; Fe, 33,000 mg; Cu, 8,800 mg; Se, 300 mg.

## Egg quality parameters

At the end of the research, the quality of the eggs was evaluated. Tree eggs were randomly chosen from each replicate, then their external and internal quality were assessed and the obtained data were pooled and then analyzed. The weight of each egg was measured, and the shape index was calculated using a caliper instrument. The shape index was calculated as a percentage using the formula (egg width/egg length). Specific gravity was determined using the method recommended by [44]. Then, each egg was broken onto a flat surface, and the height of the thick albumen and the yolk was measured using a tripod micrometer. Yolk and albumen were separated using a yolk separator and the weight of the yolk was measured using a digital scale with an accuracy of 0.01. Yolk color was assessed using the Roche Yolk color fan. Haugh units were calculated using the following formula (1):

$$\text{Haugh units (\%)}: 100 \times \log (H + 7.6 - 1.7 W 0.37) \tag{1}$$

Albumen height (H) in mm and egg weight (W) in grams.

## Blood biochemical parameters

Blood samples were collected from the brachial vein of four chickens in each treatment. Then the samples were centrifuged at a speed of 3000 rpm for 15 minutes at a temperature of 4 degrees Celsius to coagulate and blood serum was collected, then the amount of triglyceride (TG), total cholesterol (TC), albumin (AL), total protein (TP), phosphorus (P), calcium (CA), uric acid (UA), malondialdehyde (MDA) and antioxidant capacity Total (TAC) was measured in laboratory using the Lieberman-Burchard colorimetric method.

## Ileum histomorphology characteristics

At the end of experimental period (once animals reached endpoint criteria or 86 weeks of age), one randomly selected laying hen per each replicate were anaesthetized via inhalation of 60% $CO_2$ and then sacrificed by cervical dislocation.

 

CO$_2$ was delivered from compressed gas canister and delivered using a gradual fill method with a displacement rate of 60% via the chamber volume per minute using of a flowmeter. No bird died before meeting criteria for euthanasia. A sample (2 cm) was taken from the end of the ileum. After rinsing with distilled water, it was moved to the sample container containing 10% formaldehyde. The tissue processing machine used to prepare the microscopic sections was the Dided Sabz Co. Model D2080/H, manufactured in Iran. The tissue samples were embedded in histological paraffin, sliced into sections using a rotary microtome (Leica), and stained with hematoxylin and eosin. All samples were examined after preparation. The villi were evaluated based on the height, width, crypt depth, and Villus height to crypt depth ratio.

### Data analysis

The experiment was designed in a 2 × 2 factorial arrangement. The Shapiro-Wilk normality test was carried out to verify if the data was normally distributed. Then, the General Linear Model (GLM) procedure was used to analyze the data. The means comparison was followed by Duncan's test using SAS 9.4 software and level of 0.05 was considered to show significant differences among means. To perform variance analysis, the following statistical model was considered (2):

$$Y_{ijk} = \mu + a_i + b_j + (ab)_{ij} + e_{ijk} \tag{2}$$

in which $Y_{ijk}$ is the result, $\mu$ is the mean of all traits, $a_i$ is the effect of *B. coagulans*, $b_j$ is the effect of WP, $(ab)_{ij}$ is the interaction between factors, and $e_{ijk}$ is the error associated with observation. Also, the following model (3) was used to compare treatment means when an interaction effect showed to be significant:

$$Y_{ij} = \mu + t_j + e_{ij} \tag{3}$$

## Results

### Egg performance

The impact of experimental treatments on egg production, egg weight, egg mass, feed intake, and feed conversion ratio data for Lohmann LSL- lite laying hens is presented in Table 2.

In the second period (81–86 wk) and the entire period (75–86 wk), the use of *B. coagulans* alone resulted in a significant reduction compared to the control group in egg production. When *B. coagulans* mixed with WP, egg production and egg mass were not significantly different from the control group.

Feed intake was not affected by the treatments at any period of the trial. In the initial period (75–80 wk), with the use of MIX ration of WP and *B. coagulans*, FCR was reduced compared to the times when only WP was used (P < 0.05). Overall, the group receiving the MIX ration showed the best FCR, although this difference was not significant compared to the control group.

An interaction (P < 0.05) between *B. coagulans* and WP was observed for FCR as the addition of WP to *B. coagulans* improved FCR in the entire period (75–86 wk). Although, according to Table 2, this difference with other groups was not significant, the MIX ratio numerically reduced FCR.

### Egg quality parameters

The effects of dietary treatments on egg quality are shown in Table 3. No synergistic effect was found between WP and *B. coagulans* in egg quality parameters (P < 0.05). The color of the yolk was higher score in the group receiving *B. coagulans* compared to the control group. The birds that received the WP treatment did not show significant differences in the qualitative parameters studied in this experiment (P > 0.05). Eggshell weight and percentage decreased with the use of *B. coagulans* alone and in combination with WP compared to the control treatment (P < 0.05).

 

**Table 2. The effect of adding whey powder (WP), and *Bacillus coagulans* (BAC) to the diet of Lohmann laying hens on the performance in 80 - 75, 86-81, 75-86 weeks.**

| Trait | Group (Factorial interaction) | | | | SEM | Factorial main effect | | | | SEM | P-Values | | |
|---|---|---|---|---|---|---|---|---|---|---|---|---|---|
| | CON | BAC | WP | MIX | | WP (g/kg of diet) | | BAC (g/kg of diet) | | | WP | BAC | WP×BAC |
| | | | | | | – | 1 | – | 1 | | | | |
| Egg Production (%) | | | | | | | | | | | | | |
| 75-80 wk | 85.00ab | 80.56ab | 78.73b | 86.75a | 2.688 | 82.78 | 82.74 | 81.86 | 83.65 | 1.901 | 0.989 | 0.514 | 0.031 |
| 81-86 wk | 87.24a | 80.65b | 80.93b | 85.36a | 2.143 | 83.94 | 83.14 | 84.09 | 83.00 | 15.15 | 0.713 | 0.619 | 0.018 |
| 75-86 wk | 86.15a | 80.54b | 79.97b | 86.12a | 2.292 | 83.35 | 83.05 | 83.06 | 83.33 | 1.620 | 0.900 | 0.908 | 0.019 |
| Egg Weight (g/egg) (total) | | | | | | | | | | | | | |
| 75-80 wk | 59.29 | 58.84 | 58.30 | 59.27 | 0.572 | 59.06 | 58.78 | 58.80 | 59.05 | 0.405 | 0.630 | 0.656 | 0.229 |
| 81-86 wk | 61.89 | 61.21 | 62.54 | 62.06 | 0.683 | 61.55 | 62.30 | 62.21 | 61.63 | 0.483 | 0.286 | 0.406 | 0.881 |
| 75-86 wk | 60.65 | 60.06 | 61.49 | 60.67 | 0.657 | 60.35 | 61.08 | 61.07 | 60.36 | 0.464 | 0.280 | 0.297 | 0.886 |
| Egg Mass (g/d) | | | | | | | | | | | | | |
| 75-80 wk | 50.40ab | 47.45ab | 45.90b | 51.38a | 1.683 | 48.92 | 48.64 | 48.15 | 49.42 | 1.902 | 0.868 | 0.461 | 0.021 |
| 81-86 wk | 53.98a | 49.41b | 50.63ab | 52.96ab | 1.517 | 51.70 | 51.79 | 52.31 | 51.18 | 1.073 | 0.951 | 0.468 | 0.034 |
| 75-86 wk | 52.24a | 48.42b | 49.20ab | 52.23a | 1.562 | 50.33 | 50.72 | 50.72 | 50.33 | 1.104 | 0.807 | 0.803 | 0.040 |
| Feed Intake (g/d/bird) | | | | | | | | | | | | | |
| 75-80 wk | 103.44 | 99.76 | 101.42 | 101.92 | 1.949 | 101.60 | 101.67 | 102.43 | 100.84 | 0.941 | 0.97 | 0.42 | 0.295 |
| 81-86 wk | 105.71 | 101.60 | 104.08 | 104.34 | 1.651 | 103.65 | 104.21 | 104.90 | 102.97 | 0.830 | 0.74 | 0.26 | 0.200 |
| 75-86 wk | 104.68 | 100.75 | 102.87 | 103.20 | 1.684 | 102.71 | 103.04 | 103.78 | 101.98 | 0.832 | 0.85 | 0.29 | 0.220 |
| Feed Conversion Ratio (kg/kg) | | | | | | | | | | | | | |
| 75-80 wk | 2.06ab | 2.13ab | 2.21a | 1.99b | 0.104 | 2.09 | 2.10 | 2.14 | 2.06 | 0.031 | 0.93 | 0.20 | 0.017 |
| 81-86 wk | 1.96 | 2.07 | 2.06 | 1.97 | 0.049 | 2.02 | 2.02 | 2.01 | 2.02 | 0.030 | 0.95 | 0.84 | 0.054 |
| 75-86 wk | 2.01 | 2.10 | 2.10 | 1.98 | 0/050 | 2.05 | 2.04 | 2.05 | 2.04 | 0.104 | 0.76 | 0.81 | 0.045 |

CON = basal diet (without additive); WP: whey powder; BAC: *Bacillus coagulans* ($4 \times 10^6$); MIX = basal diet + WP and BAC.

Different letters (a-b) within a row represent significant differences (P < 0.05) among treatments means.

## Blood biochemical Parameters

The results of dietary treatments on blood biochemical parameters are presented in Table 4. No significant synergistic effect was observed between WP and *B. coagulans* in blood biochemical parameters, other than malondialdehyde (P < 0.05). According to Fig 1, the use of WP along with *B. coagulans* resulted in reduced levels of malondialdehyde in the serum compared to when WP was used alone (P < 0.05).

## Ileum histomorphology parameters

The main effects of the experimental factors (adding WP and *B. coagulans* bacteria in the diet) and their interaction effects on the histomorphological parameters of the ileum of Lohmann LSL-lite laying hens are shown in Table 5. As shown in Fig 2, the width of the ileum villi was not significantly affected by the WP, but it increased significantly due to the addition of *B. coagulans* to the diet (P ≤ 0.05). WP did not make a significant difference on any of the histomorphological parameters investigated in this experiment (P < 0.05). The interaction effect of WP and *B. coagulans* in the experimental groups did not have a significant effect on any of the ileum histomorphology parameters investigated in this study (villus height, villus width, crypt depth and the villus height to crypt depth ratio in laying hens) (P > 0.05).

**Table 3. The effect of adding whey powder (WP) and *Bacillus coagulans* (BAC) to the diet of Lohmann laying hens on egg quality parameters (86 weeks).**

| Trait | Group (Factorial interaction) | | | | SEM | Factorial main effect | | | | SEM | P-Values | | |
|---|---|---|---|---|---|---|---|---|---|---|---|---|---|
| | | | | | | WP (g/kg of diet) | | BAC (g/kg of diet) | | | | | |
| | CON | BAC | WP | MIX | | – | 1 | – | 1 | | WP | BAC | WP×BAC |
| Egg weight (Sample) | 63.22 | 61.17 | 66.33 | 64.90 | 1.786 | 62.19 | 65.62 | 64.78 | 63.06 | 1.263 | 0.070 | 0.341 | 0.865 |
| Egg length(cm) | 5.79 | 5.88 | 6.00 | 6.02 | 0.127 | 5.84 | 6.01 | 5.90 | 5.95 | 0.090 | 0.194 | 0.693 | 0.811 |
| Egg width (cm) | 4.40 | 4.35 | 4.49 | 4.44 | 0.049 | 4.37 | 4.46 | 4.44 | 4.39 | 0.035 | 0.095 | 0.344 | 0.960 |
| Shape index | 76.25 | 74.12 | 74.88 | 73.77 | 1.569 | 75.18 | 74.32 | 75.56 | 73.94 | 1.110 | 0.588 | 0.315 | 0.749 |
| Albumen weight (g) | 39.23 | 37.69 | 41.80 | 40.67 | 1.641 | 38.46 | 41.23 | 40.52 | 39.18 | 0.424 | 0.107 | 0.424 | 0.901 |
| yolk weight (g) | 17.72 | 18.04 | 18.20 | 18.43 | 0.616 | 17.88 | 18.32 | 17.96 | 18.24 | 0.435 | 0.485 | 0.656 | 0.943 |
| Shell weight (g) | 6.27[a] | 5.44[b] | 6.33[a] | 5.80[ab] | 0.175 | 5.85[b] | 6.07[a] | 6.30 | 5.62 | 0.124 | 0.237 | 0.009 | 0.412 |
| yolk height(mm) | 16.88 | 17.71 | 17.31 | 17.80 | 0.359 | 17.29 | 17.56 | 17.10 | 17.75 | 0.254 | 0.469 | 0.082 | 0.634 |
| Albumen height (mm) | 5.91 | 6.22 | 6.65 | 5.81 | 0.410 | 6.06 | 6.23 | 6.28 | 6.02 | 0.290 | 0.690 | 0.532 | 0.176 |
| Haugh unit | 74.43 | 77.06 | 79.36 | 73.16 | 3.054 | 75.74 | 76.26 | 76.89 | 75.11 | 2.160 | 0.868 | 0.566 | 0.163 |
| Egg yolk index | 0.37 | 0.40 | 0.38 | 0.39 | 0.009 | 0.38 | 0.39 | 0.38 | 0.39 | 0.006 | 0.652 | 0.097 | 0.529 |
| yolk color | 4.5[b] | 5.67[a] | 5.17[ab] | 5.83[a] | 0.266 | 5.08 | 5.50 | 4.83[b] | 5.75[a] | 0.188 | 0.133 | 0.003 | 0.359 |
| shell strength | 28.05 | 29.68 | 27.46 | 28.45 | 1.039 | 41.51 | 43.59 | 41.68 | 43.42 | 2.55 | 0.570 | 0.636 | 0.866 |
| Albumen (%) | 62.02 | 61.68 | 62.99 | 62.61 | 1.135 | 61.69 | 62.80 | 62.50 | 61.99 | 0.802 | 0.346 | 0.656 | 0.908 |
| yolk (%) | 28.05 | 29.68 | 27.46 | 28.45 | 1.039 | 28.86 | 27.95 | 27.75 | 29.06 | 0.735 | 0.393 | 0.222 | 0.760 |
| Shell (%) | 9.93[a] | 8.95[b] | 9.55[ab] | 8.95[b] | 0.292 | 9.44 | 9.25 | 9.74[a] | 8.95[b] | 0.206 | 0.526 | 0.013 | 0.526 |
| Specific gravity | 1.081 | 1.083 | 1.084 | 1.086 | 0.002 | 1.082 | 1.085 | 1.083 | 1.084 | 0.002 | 0.135 | 0.445 | 1.000 |

CON = basal diet (without additive); WP: whey powder; BAC: *Bacillus coagulans* ($4 \times 10^6$); MIX = basal diet + WP and BAC.

Different letters (a-b) within a row represent significant differences (P < 0.05) among treatments means.

**Table 4. The effect of adding whey powder (WP) and *Bacillus coagulans* (BAC) to the diet of Lohmann laying hens on blood biochemical parameters (86 weeks).**

| Trait | Group (Factorial interaction) | | | | SEM | Factorial main effect | | | | SEM | P-Values | | |
|---|---|---|---|---|---|---|---|---|---|---|---|---|---|
| | CON | WP | BAC | MIX | | WP (g/kg of diet) | | BAC (g/kg of diet) | | | WP | BAC | WP×BAC |
| | | | | | | – | 1 | – | 1 | | | | |
| CHO (mg/dl) | 107.25 | 119.00 | 115.75 | 126.50 | 16.169 | 113.13 | 121.13 | 111.50 | 122.75 | 11.433 | 0.63 | 0.50 | 0.97 |
| TG (mg/dl) | 1545.00 | 2097.50 | 1795.00 | 1762.50 | 452.367 | 1821.25 | 1778.75 | 1670.00 | 1930.00 | 319.872 | 0.92 | 0.57 | 0.53 |
| TP (g/dl) | 6.28 | 6.18 | 6.05 | 6.00 | 0.358 | 6.23 | 6.03 | 6.16 | 6.09 | 0.253 | 0.58 | 0.83 | 0.94 |
| P (mg/dl) | 7.18 | 6.58 | 6.53 | 5.20 | 1.214 | 6.88 | 5.86 | 6.85 | 5.89 | 0.859 | 0.42 | 0.44 | 0.77 |
| CA (mg/dl) | 32.20 | 36.83 | 26.18 | 26.63 | 4.774 | 34.51 | 26.40 | 29.19 | 31.73 | 3.376 | 0.11 | 0.60 | 0.67 |
| AL (g/dl) | 2.48 | 2.20 | 2.20 | 2.15 | 0.148 | 2.34 | 2.18 | 2.34 | 2.18 | 0.105 | 0.29 | 0.29 | 0.46 |
| UA (mg/dl) | 2.47 | 3.92 | 2.94 | 2.90 | 0.599 | 3.20 | 2.92 | 2.70 | 3.41 | 0.424 | 0.64 | 0.25 | 0.24 |
| MDA (nmol/ml) | 2.55[b] | 3.55[ab] | 5.83[a] | 2.70[b] | 0.918 | 3.05 | 4.26 | 4.19 | 3.125 | 0.649 | 0.21 | 0.27 | 0.04 |
| TAC (mmol/liter) | 0.92 | 1.18 | 0.93 | 0.92 | 0.126 | 1.05 | 0.92 | 0.93 | 1.05 | 0.089 | 0.32 | 0.35 | 0.28 |

CON = basal diet (without additive); WP: whey powder; BAC: *Bacillus coagulans* ($4 \times 10^6$); MIX = basal diet + WP and BAC; CHO: Cholesterol; TG: tri-glyceride; TP: total protein; P: phosphorus; CA: calcium; UA: uric acid; MDA: Malondialdehyde; TAC: total antioxidant capacity.

Different letters (a-b) within a row represent significant differences (P < 0.05) among means in main effects and significant interactions

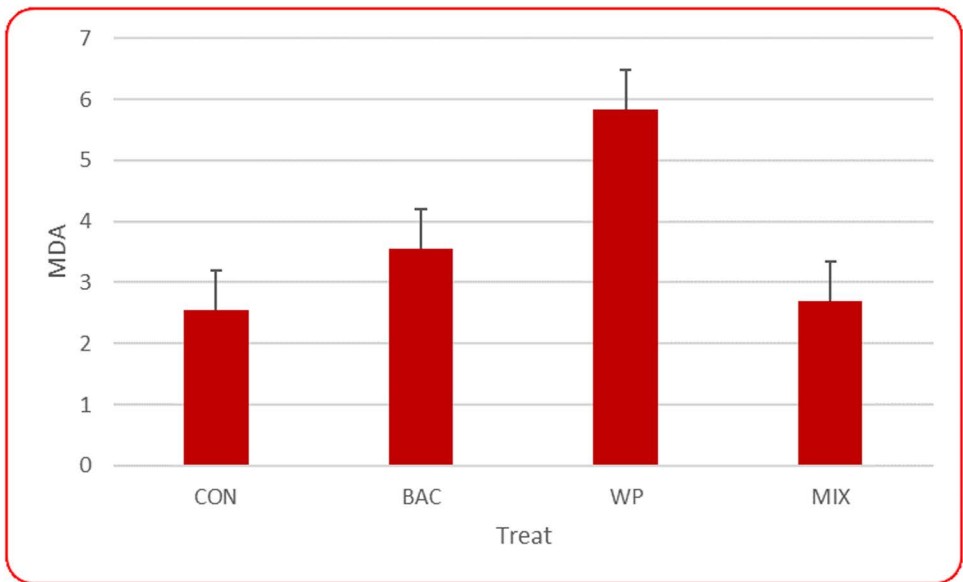

**Fig 1. Average concentration of malondialdehyde in serum samples in the different groups.**

**Table 5. The effect of adding whey powder (WP) and *Bacillus coagulans* (BAC) to the diet of Lohmann laying hens on ileum histomorphology parameters (86 weeks).**

| Trait | Group (Factorial interaction) | | | | SEM | Factorial main effect | | | | SEM | P-Value | | |
|---|---|---|---|---|---|---|---|---|---|---|---|---|---|
| | CON | BAC | WP | MIX | | WP (g/kg of diet) | | BAC (g/kg of diet) | | | WP | BAC | WP×BAC |
| | | | | | | – | 1 | – | 1 | | | | |
| Villus height (μm) | 479.83 | 528.76 | 526.68 | 513.05 | 19.32 | 504.30 | 519.88 | 503.25 | 520.90 | 5.466 | 0.444 | 0.388 | 0.144 |
| Villus width (μm) | 97.40 | 182.68 | 98.51 | 102.05 | 38.410 | 102.55 | 100.28 | 97.95[b] | 104.88[a] | 0.597 | 0.315 | 0.011 | 0.150 |
| Crypt depth (μm) | 99.61 | 108.63 | 103.63 | 103.50 | 2.761 | 103.98 | 103.58 | 101.63 | 105.90 | 0.781 | 0.899 | 0.160 | 0.149 |
| Villus height/ Crypt depth | 4.83 | 4.88 | 5.08 | 4.96 | 0.189 | 4.86 | 5.02 | 4.96 | 4.92 | 0.133 | 0.422 | 0.849 | 0.663 |

CON = basal diet (without additive); WP: whey powder; BAC: *Bacillus coagulans* ($4 \times 10^6$); MIX = basal diet + WP and BAC.

Different letters (a-b) within a row represent significant differences (P < 0.05) among means in main effects and significant interactions.

## Discussion

### Egg performance

In the second period (81–86 wk) and the entire period (75–86 wk), the use of *B. coagulans* alone resulted in a significant reduction compared to the control group in egg production. When *B. coagulans* mixed with WP, egg production and egg mass were not significantly different from the control group. This effect is likely due to the improvement of the gut microbiota by *B. coagulans* and the inhibition of harmful bacteria. The increase in the width of the ileum villi also indicates better absorption of nutrients, especially the highly digestible protein and fat of WP and its soluble vitamins [45], so the MIX diet can be effective in improving performance compared to the use of either alone.

The effect of *B. coagulans* on production performance in aged laying hens is unclear. However, Wang et al. [46] indicated that *B. coagulans* improves production performance and host health in late-laying period. The simultaneous addition of WP and *Pediococcus acidilactici* in the diet of chickens during the final period of production did not improve production performance [47]. However, in the present experiment, when WP was administered alongside *B. coagulans*, production

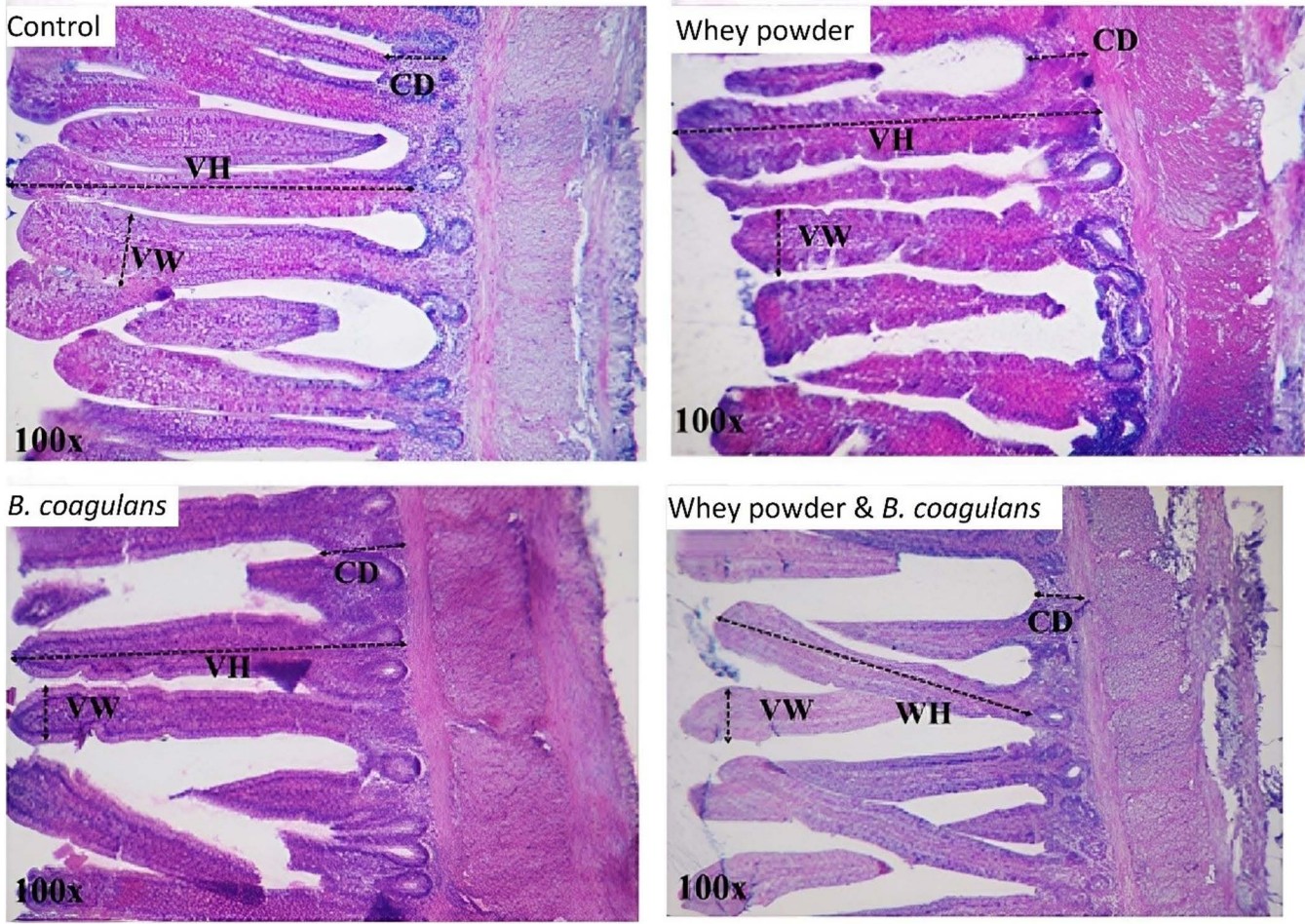

**Fig 2. Photomicrographs of haematoxylin and eosin staining of ileal tissues different treatments in Lohmann LSL-lite laying hens at the end of 86 weeks.** Magnification was 100 x objective lens VH = villus height; VW = villus width; CD = crypt depth.

performance improved. This improvement may be related to the type of probiotic consumed and its potential synergistic effect. Also, in a research conducted by Guanrun et al. [48] with three different concentrations of *B. coagulans* in laying hens at the end of the laying period, it was found that *B. coagulans* significantly improves the production performance of laying hens at the end of the laying period.

In a study by Xu et al. [36] in laying hens during the peak production period, supplementation with *B. coagulans* X26 resulted in increased egg weight and egg white protein, and decreased feed intake and feed-to-egg ratio, however, egg weight was not affected by any of the treatments in this experiment. Moreover, in the experiment Wang et al. [46], with an increase in the level of *B. coagulans*, the FCR decreased significantly (P<0.05). At the same time, it did not affect feed consumption, which was similar to our results. Additionally, Fitzpatrick [49] observed that *B. coagulans*, as a probiotic, increases the parameters related to the growth of chickens. In another study conducted by Zhou et al. [32] with *B. coagulans* as a probiotic in Guangxi yellow chickens, a lower FCR and a higher survival rate were observed, and probiotic administration through the basal diet had beneficial effects on final weight and BWG. Data show that administration of *B. coagulans* ATCC 7050 improves FCR in broiler chickens probably through improving the balance of gut microflora [33]. Pedroso et al. [50] exhibited that changing the composition of the intestinal bacterial community is responsible for better

growth in chickens; Therefore, *B. coagulans* may improve dietary FCR by increasing lactobacilli and decreasing coliform bacterial populations in the duodenum. Lactose in WP acts as an energetic substrate for *Lactobacillus spp.* [10], which may suppress total coliforms and *E. coli* through competitive exclusion by increasing *Lactobacillus spp.* levels [51]. In this study, *B. coagulans* also improved intestinal function, and the simultaneous use of the two seems (MIX) to provide better performance than the use of either one alone.

In this research, even with the high age of chickens participating in the experiment, this effect was observed and FCR improved, although this difference was not significant.

However, it is very difficult to directly evaluate all the different studies that have used probiotics, because the effectiveness of the use of a probiotic depends on many factors, including age, genetics, type of probiotic, etc. [50]. In the research conducted with the addition of WP in the diets of broilers, it was observed that the addition of WP in the diet increases growth performance [37,47,52]. In another experiment conducted by Jazi et al. [53] by adding WP and *B. subtilis* to Japanese quail diets, it was also observed that WP and *B. subtilis* and the mixture of them had higher BWG and lower FCR than the control group during the growth period. Greenhalgh et al. [54] also observed that adding WP to corn-based diets improved FCR. On the other hand, BWG and FCR in birds fed with experimental diets were significantly better than the control diet for the entire growth period, however, the results of this study show the synergistic effects of *B. subtilis* and WP supplementation compared to their application alone, did not confirm [55].

It has been reported that the use of whey did not affect FCR, ADFC, EPEF, and mortality [56]. On the other hand, in a study conducted by Zarei et al. [57] on broilers using a type of probiotic with WP. The test results showed that the lowest feed intake in the control group was without significant difference from the probiotic group. These results were consistent with the results of this research. Samli et al. [58] noted the improvement of FCR in broilers receiving probiotics and WP. However, Rastad et al. [59] announced that no significant difference with the control group in broilers that received WP mixed with probiotics [60]. Furthermore, Samli et al. [58] displayed that no improvement in FCR after adding WP. Some researchers also reported that the mixture of WP and *Pediococcus acidilactici* leads to a synergistic effect that leads to positive modulation of cecal bacteria and better performance [47]. The possible reasons for the difference in these results may be due to the difference in the type, species, survival power of the species, administration level, application method, application frequency, general diet, bird age, genetics, general health of the farm and environmental stress factors, management method, etc.

## Egg quality parameters

According to an experiment conducted by Bouassi et al. [61] in laying hens, Liquid whey (LW) did not affect the quality parameters of the eggs, and the body weight of the birds at the first laying, egg weight, and quality traits were not affected by the dose of LW. In this trial, none of the quality indicators of eggs were affected by the interaction between WP and *B. coagulans*, which requires more studies in this field.

The results of the experiments by Wang et al. [46] showed that the Albumen height, Haugh unit, egg yolk color, eggshell strength was not affected by the level of *B. coagulans*. The yolk color index showed a higher score; which is consistent with the results reported by Zeweil et al. [62] and the findings of the present study. The exact mechanisms by which probiotics affect yolk color are still being investigated, but there are several theories. Probiotics may increase the absorption of carotenoids, the yellow pigments of egg yolks, from the hen's diet [63]. Probiotic bacteria also produce enzymes that help break down complex feed molecules, such as carotenoid esters, making them more readily available for absorption [64]. In this study, the weight and percentage of eggshell decreased, contrary to many previous studies [46,65,66], without providing a scientific reason for this. These conflicting results indicate that the effect of *B. coagulans* on egg quality is uncertain and depends on various factors such as bacterial strains, dose, and animal growth period.

## Blood biochemical parameters

In the work of Wang et al. [46] in laying hens, it shows that serum total cholesterol, triglyceride, phosphorus, and uric acid were not affected by the diet containing *B. coagulans*, which was consistent with the results of the experiments conducted in this study. The research of some researchers showed that the total protein content of poultry blood increases with age as a result of metabolic changes in the animal and reflects various disorders in nutritional characteristics caused by insufficient or excessive intake of protein in feed mixtures [67], due to the high age. The birds participating in this experiment (75–86 weeks old) and the normality of the albumin level in this research indicate the adequacy of the protein source and the favorable breeding environment.

*B. coagulans* could alleviate oxidative stress by increasing the activities of myeloperoxidase [68] and anti-superoxide anion free radical (AFASER), decreasing the content of malondialdehyde (MDA), regulating the transcriptional regulation levels of antioxidant enzymes and Nrf2-Keap1 signaling molecules [69]. Wang et al. [46] exhibited that the diet containing *B. coagulans* did not affect the serum concentration of MDA or the activities of T-AOC, SOD, and CAT. On the other hand Zhang et al. [70] announced that MDA content was significantly decreased in broilers affected by *B. coagulans*. In addition, Liu et al. [71] showed that MDA is the main product of lipid peroxide degradation, which reflects the intensity and speed of lipid peroxide formation, as well as the rate of lipid peroxidation and free radical attack in cells. Zhang et al. [70] observed a decrease in MDA with the addition of *B. coagulans*, implying that enhance of *B. coagulans* effectively reduces lipid peroxidation in broilers. On the other hand Jazi et al. [53] disclosed that the addition of WP and *B. subtilis* to the diets of Japanese quails led to a decrease in MDA levels in the thigh muscle, although it increased the activities of SOD and GSH-PX, which is the same as our results for laying hens. These results indicate that WP increases the production of antioxidant agents, which is consistent with the findings of El-Desouky et al. [72]. It was also observed in experiments Jazi et al. [53] that WP and *B. subtilis* and their mixture decreased the serum cholesterol concentration in birds fed experimental diets compared to the control group, which was in contrast to our results for laying hens. Also, in parallel with our work, no difference was observed in the serum level of triglyceride and HDL-C in Japanese quails studied by Jazi et al. [53].

## Ileum histomorphology parameters

The main function of the small intestine mucosa, which consists of a layer of epithelial cells, is to digest and absorb nutrients and inhibit pathogenic bacteria and toxic substances in the intestinal lumen [73]. A crypt is a tubular gland formed by the epithelium of the small intestine that protrudes into the lamina propria at the root of the villi [74]. A higher ratio of villus height to crypt depth (VH/CD) indicates a higher rate of digestion and absorption performance [75]. Also, longer villi, shallower crypt, and the ratio of villi height to crypt depth indicate that intestinal mucosal cells are more mature, more complete, and have a stronger intestinal digestion and absorption ability [76]. Consistent with the present study, in a research conducted in laying hens at the end of the laying period, *B. coagulans* feed additive had positive effects on some morphological characteristics of villi in the jejunum and ileum [46]. In this experiment, the use of *B. coagulans* alone caused a significant increase in the width of the ileum villi. On the other hand, the results of another experiment showed that *B. coagulans* X26 significantly increases the height of intestinal villi, and the ratio of villus height to crypt depth in laying hens [36]. In broiler chickens, no significant difference was observed between the experimental groups when using *B. coagulans* regarding villus length, crypt depth and villus length: crypt depth ratio in the intestine [33]. Researchers conducted an experiment on broiler chickens using dry whey powder and calcium butyrate as supplements to their diet. Results showed that adding either supplement led to an increase in the height of the villi or the ratio of the villus length to the crypt depth in the duodenum compared to the control group [37] . However, WP did not affect the histomorphological parameters of the ileum in this study.

## Conclusions

The experiment revealed no synergistic effect between WP and *B. coagulans* on egg quality and ileum histomorphology parameters. However, MIX dietary improved production performance and reduced serum malondialdehyde levels in

Lohmann laying hens during late production compared to individual use. This MIX improved FCR, which is suggested to be further investigated in other species including broilers.

## Author contributions

**Conceptualization:** Mehran Torki, Alireza Abdolmohammadi.

**Data curation:** Zahra Hamzehee, Khodabakhsh Rashidi, Alireza Abdolmohammadi.

**Formal analysis:** Zahra Hamzehee, Alireza Abdolmohammadi.

**Investigation:** Zahra Hamzehee, Mehran Torki, Khodabakhsh Rashidi, Alireza Abdolmohammadi.

**Methodology:** Zahra Hamzehee, Mehran Torki, Khodabakhsh Rashidi, Alireza Abdolmohammadi.

**Project administration:** Mehran Torki.

**Resources:** Mehran Torki.

**Software:** Zahra Hamzehee.

**Supervision:** Mehran Torki.

**Validation:** Mehran Torki, Khodabakhsh Rashidi.

**Visualization:** Khodabakhsh Rashidi.

**Writing – original draft:** Zahra Hamzehee.

**Writing – review & editing:** Mehran Torki.

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
