## [Decision Letter · Decision Letter 0]

20 Feb 2025

PONE-D-24-54255Effects of dietary of Bacillus coagulans, whey powder, and their interaction on the performance of Lohmann LSL-lite laying hens in the late production phasePLOS ONE

Dear Dr. Torki,

Thank you for submitting your manuscript to PLOS ONE. After careful consideration, we feel that it has merit but does not fully meet PLOS ONE’s publication criteria as it currently stands. Therefore, we invite you to submit a revised version of the manuscript that addresses the points raised during the review process.

We look forward to receiving your revised manuscript.

Kind regards,

Shengqian Sun

Academic Editor

PLOS ONE

2. Please amend your list of authors on the manuscript to ensure that each author is linked to an affiliation. Authors’ affiliations should reflect the institution where the work was done (if authors moved subsequently, you can also list the new affiliation stating “current affiliation:….” as necessary).

**Comments to the Author**

1. Is the manuscript technically sound, and do the data support the conclusions?

Reviewer #1: Partly

Reviewer #2: Partly

Reviewer #3: Yes

2. Has the statistical analysis been performed appropriately and rigorously? 

Reviewer #1: No

Reviewer #2: Yes

Reviewer #3: Yes

3. Have the authors made all data underlying the findings in their manuscript fully available?

Reviewer #1: Yes

Reviewer #2: Yes

Reviewer #3: Yes

4. Is the manuscript presented in an intelligible fashion and written in standard English?

Reviewer #1: Yes

Reviewer #2: No

Reviewer #3: No

5. Review Comments to the Author

Reviewer #1: Abstract

line 5. ... to produce high-quality ...

line. correct 106 CFU

Add some factual results in the abstract from the results.

Introduction.

Line 26-27. Please revise. The ban of antibiotics is already a subject treated in many papers. Il will be better to revise the sentence and revised it in a broader content, for example, the ban of antibiotics usage in EU, USA, China, and Brazil. When they have been banned.

Line 28, provide different reference for each alternative provided: for example: development of antibiotic substitutes such as probiotics and postbiotics (https://doi.org/10.3390/ani11123431), botanical plants (https://doi.org/10.3382/ps/pev014) waste materials such as by-products, oilseeds etc ..(https://doi.org/10.3390/agriculture14101811), enzymes (https://doi.org/10.1016/j.scitotenv.2024.171757), and so on. Also, mention some effect found, as not all of them have been reported to be suitable.

In the introduction section there is no word about the need to use these dietary ingredients in aged laying hens. Please make some references to this aspect. maybe some comparison with young (first phase of laying) laying hens. Provide a wider background of your study.

line 63-69. There is no word about the late laying phase in the aim of the study.

Material and methods.

The total number of laying hens was 144, which mean 24 per group, and 4 repetitions per each group. Using only 24 hens for a group, is a very low number to draw significant conclusion. The authors should have used at least 30 hens per group for a better robustness of the statistical data in terms of production performances.

Table 1, please provide the ingredients as 100% ingredients. Also, the nutritional/chemical composition of the diets should be provided (CP, EE, CF, ME).

Line 92. why such high temperature?

Line 101. 6 eggs per group? or average egg samples. This is also a very low number of observations to draw meaningful conclusions.

Line 148-153, please revise. Use Capitals when a different sentence begin.

Table 2 and 3 does not make any sense. In Table 2 the average values are provided, ans in table 3 are different average values, but not for all parameters.

Same observation for Table 4 and 5.

In Table 6, the egg weight and shell weight should be provided. The 6 eggs taken into analyses have different weight compared to those presented in the above tables.

Table 6 an 7, the MIX group is missing from the provided results, but is included in the statistical model.

Table 8. the significance is missing, where is the p value presented? Better provide MDA results as a graphical representation.

Discussion

For production performances, the discussion is very scarce, as this aspect is very important for late laying hens production.

Same for egg quality parameters.

Conclusion,

Revise it with some results from this study, not with an assumption.

Figure 1, is labeled as T1, T2, T3 and T4, please revise it according to the groups presented in this study.

Reviewer #2: The manuscript entitled "Effects of Dietary Bacillus coagulans, Whey Powder, and Their Interaction on the Performance of Lohmann LSL-Lite Laying Hens in the Late Production Phase" demonstrates sufficient quality for acceptance into the journal; however, several questions must be addressed prior to publication.

In Table 1, "Composition and Content of Nutrients," the specific nutrients are not clearly delineated.

The justification for the amount of B. coagulans (4 × 10^6 CFU) utilized in the study should be provided with appropriate citations.

Table 1 indicates that the mineral and vitamin premix is not formulated for laying hens.

The formula used for calculating the Feed Conversion Ratio (kg/kg) appears to be erroneous, as the Feed Conversion Ratio is a unitless measure.

Analyses presented in similar tables warrant re-evaluation; for instance, in Table 3, the main and interaction effects analysis is not adequately represented.

In Table 4, the calculation of the conversion factor is incorrect; the feed conversion factor for hens of this age is estimated to be approximately 2, which seems implausible.

Given their relevance, the following articles should be incorporated into the introduction and discussion sections:

Reviewer #3: Effects of dietary of Bacillus coagulans, whey powder, and their interaction on the performance of Lohmann LSL-lite laying hens in the late production phase

L11: fix the number (4*106 CFU) here and everywhere in the manuscript.

L17-18: revise, Furthermore, FCR in the first (75-80 weeks) was affected by the decreased effect of WP and B.

L19: revise, color of the yolk was thicker.

Update reference list in the introduction to include more recent papers. DOI: 10.1080/1828051X.2024.2418409

Revise the introduction.

The birds were giving the treatments at 1 d old or at different age? provide more details

Table 1: provide the chemical composition of the diet. The diet was formulated according to which guideline?

L102-103: 6 eggs were randomly chosen from each treatment in the experiment, and their external and internal quality were assessed. This is an extremely low number of eggs.

Table 2 and 3: all significant 2 ways interactions need to be discussed in the result section.

From table 3: it seems that control was the best treatment, the additives lowered the performance.

What is the difference between data in table5 and table 4?

L225-230: revise this part to reflect the results you got from the experiment.

6. PLOS authors have the option to publish the peer review history of their article (what does this mean? ). If published, this will include your full peer review and any attached files.

**Do you want your identity to be public for this peer review?** For information about this choice, including consent withdrawal, please see our Privacy Policy .

Reviewer #1: No

Reviewer #2: No

Reviewer #3: **Yes: ** Ala Abudabos

---

## [Author Response · Author response to Decision Letter 1]

20 Mar 2025

Response to Reviewers

Date: 03/19/2025

PLOS ONE

Subject: Response to Reviewer Comments on Manuscript entitled "Effects of dietary of Bacillus coagulans, whey powder, and their interaction on the performance of Lohmann LSL-lite laying hens in the late production phase"

Dear Editors and Reviewers,

We would like to express our sincere gratitude for the time and effort you dedicated to reviewing our manuscript. We greatly appreciate your constructive comments, which have helped improve the quality of our paper. We have carefully addressed all of the reviewer comments, and below, we provide detailed responses to each point raised. Thank you for your valuable feedback and suggestions. We have addressed all your questions and concerns in the revised manuscript and carefully incorporated your recommendations to reflect these changes. It would be very kind of you to inform us in the case of any neglected comment!

Reviewer #1:

Abstract: line 5. ... to produce high-quality ...

line. correct 106 CFU

Thank you for your comment. All text has been fixed.

Add some factual results in the abstract from the results.

We thank the reviewer for this comment. It was added.

Introduction: Line 26-27. Please revise. The ban of antibiotics is already a subject treated in many papers. Il will be better to revise the sentence and revised it in a broader content, for example, the ban of antibiotics usage in EU, USA, China, and Brazil. When they have been banned.

Thank you for your comment. The mentioned points were done.

Line 28, provide different reference for each alternative provided: for example: development of antibiotic substitutes such as probiotics and postbiotics (https://doi.org/10.3390/ani11123431), botanical plants (https://doi.org/10.3382/ps/pev014) waste materials such as by-products, oilseeds etc. (https://doi.org/10.3390/agriculture14101811), enzymes (https://doi.org/10.1016/j.scitotenv.2024.171757), and so on. Also, mention some effect found, as not all of them have been reported to be suitable.

Thank you for the helpful suggestions. We have incorporated the suggested references to further support this statement [https://doi.org/10.3390/ani11123431], [https://doi.org/10.3382/ps/pev014], [https://doi.org/10.1016/j.scitotenv.2024.171757], [https://doi.org/10.3390/agriculture14101811]. Additionally, new references were added [https://doi.org/10.1080/1828051X.2024.2418409],[https://doi.org/10.1016/j.livsci.2021.104806], and etc.

In the introduction section there is no word about the need to use these dietary ingredients in aged laying hens. Please make some references to this aspect. maybe some comparison with young (first phase of laying) laying hens. Provide a wider background of your study.

We really appreciate your attention. The mentioned points have been added.

line 63-69. There is no word about the late laying phase in the aim of the study.

Thank you for raising this important point. It was added.

Material and methods.

The total number of laying hens was 144, which mean 24 per group, and 4 repetitions per each group. Using only 24 hens for a group, is a very low number to draw significant conclusion. The authors should have used at least 30 hens per group for a better robustness of the statistical data in terms of production performances.

We appreciate your feedback. However, we would like to clarify that our study included 6 replicates per group, not 4 as mentioned in the comment. This design aligns with standard methodologies used in poultry nutrition research. While we acknowledge the suggestion to increase the number of birds per group for enhanced statistical robustness, we believe that the current design provides reliable data for production performance evaluation. Nonetheless, we will consider this recommendation for future studies.

Table 1, please provide the ingredients as 100% ingredients. Also, the nutritional/chemical composition of the diets should be provided (CP, EE, CF, ME).

Thank you for your insightful comment. The points mentioned have been corrected as 100% ingredients and added.

Line 92. why such high temperature?

Thank you for your question. In this experiment, we tried to maintain the temperature at 20°C, but given that almost half of our experiment was conducted in the summer (our study was conducted from July to November), sometimes the temperature reached 22°C.

https://lohmann-breeders.com/media/2021/03/LTZ_MG_management-systems_EN.pdf#page=10.15

page 13

Line 101. 6 eggs per group? or average egg samples. This is also a very low number of observations to draw meaningful conclusions.

Thank you for highlighting that. We realize the manuscript lacked clarity. To assess quality parameters, we used 3 eggs per replicate and pooled the data for analysis. A total of 18 samples per group were evaluated.

Line 148-153, please revise. Use Capitals when a different sentence begin.

Thank you for your attention, all text has been corrected.

Table 2 and 3 does not make any sense. In Table 2 the average values are provided, ans in table 3 are different average values, but not for all parameters. Same observation for Table 4 and 5.

Thank you for your valuable insights; all the tables have been revised accordingly. Before updating my manuscript, we used the slice method to compare means after establishing the significance of the interaction effect, Considering the important points you taught us. The table in the paper below has been used as a template for revising the corrected tables. Thank you for enhancing our study.

[https://doi.org/10.1016/j.psj.2021.101553]

In Table 6, the egg weight and shell weight should be provided.

Thank you for your feedback. I have now added the eggshell weight and percentage to the table

The 6 eggs taken into analyses have different weight compared to those presented in the above tables.

Thank you for your feedback. As you know, the egg weights in the performance table reflect data collected through daily sampling over the entire period. In contrast, the egg weights in the quality parameter table are derived from three pooled eggs per repetition, totaling 18 eggs per group, and were measured at the end of the period. As a result, the numbers in these two tables differs.

Table 6 and 7, the MIX group is missing from the provided results, but is included in the statistical model.

Thank you for raising this important point.

Table 8. the significance is missing, where is the p value presented? Better provide MDA results as a graphical representation.

All tables were revised.

Discussion

For production performances, the discussion is very scarce, as this aspect is very important for late laying hens production. Same for egg quality parameters.

Thank you for your valuable feedback. We have addressed your suggestion.

Conclusion: Revise it with some results from this study, not with an assumption.

We thank the reviewer for this comment. It was revised.

Figure 1, is labeled as T1, T2, T3 and T4, please revise it according to the groups presented in this study.

Thank you for your comment. I have revised Figure 1 and updated the labels to match the groups presented in this study.

Reviewer #2:

In Table 1, "Composition and Content of Nutrients," the specific nutrients are not clearly delineated.

Thank you for your valuable feedback. We have revised Table 1 to clearly delineate the specific nutrients, ensuring better clarity and readability. We appreciate your insightful suggestion.

The justification for the amount of B. coagulans (4 × 106 CFU) utilized in the study should be provided with appropriate citations.

Our decision to use this B. coagulans count was based on previous studies and the suggestion of Parcilact (Parcilact co. Shiraz, Iran), as well as the count that we had available.

http://dx.doi.org/10.3382/ps.2009-00319

https://doi.org/10.1080/1828051X.2022.2163931

https://doi.org/10.1016/j.psj.2022.101835

Table 1 indicates that the mineral and vitamin premix is not formulated for laying hens.

Response: Yes, you’re right. It was actually a mistake, which was corrected in the revised version.

The formula used for calculating the Feed Conversion Ratio (kg/kg) appears to be erroneous, as the Feed Conversion Ratio is a unitless measure.

As you know, FCR was calculated based on kilograms of feed consumed per kilogram of eggs produced. In the following articles, FCR is also used as a unit.

https://doi.org/10.1016/j.psj.2024.104013 (kg/kg in 2024 poultry science)

https://sau.edu.bd/uploads/pdf/dept/class_lecture/24_2020-08-11_5f32fee3c41d6.pdf

Analyses presented in similar tables warrant re-evaluation; for instance, in Table 3, the main and interaction effects analysis is not adequately represented.

Thank you for your valuable insights; all the tables have been revised accordingly. Before updating my article, I used the slice method to compare means after establishing the significance of the interaction effect, Considering the important points you taught me. The table in the article below has been used as a template for revising the corrected tables. Thank you for enhancing our study.

https://doi.org/10.1016/j.psj.2021.101553

In Table 4, the calculation of the conversion factor is incorrect; the feed conversion factor for hens of this age is estimated to be approximately 2, which seems implausible.

The values were also double-checked, and the results are the same as previously reported.

Reviewer #3:

L11: fix the number (4*106 CFU) here and everywhere in the manuscript.

I am sincerely grateful. Fixed throughout the manuscript.

L17-18: revise, Furthermore, FCR in the first (75-80 weeks) was affected by the decreased effect of WP and B.

We thank the reviewer for their thoughtful suggestion. Furthermore, FCR was reduced in the first period (75–80 weeks) under the influence of the mixed group compared to the control group or when used alone.

L19: revise, color of the yolk was thicker.

It was corrected.

Update reference list in the introduction to include more recent papers. DOI: 10.1080/1828051X.2024.2418409

Thank you for the helpful suggestions. We have added the suggested reference to support this point [DOI: 10.1080/1828051X.2024.2418409].

Revise the introduction.

We appreciate for raising this important point. It was revised.

The birds were giving the treatments at 1 d old or at different age? provide more details

Thank you for your question. Birds were assessed for performance, quality, blood, and histomorphology parameters at the end of the production period, specifically from the beginning of week 75 to the end of week 86.

Table 1: provide the chemical composition of the diet. The diet was formulated according to which guideline? It was based on the Lohmann LSL Lite rearing catalogue with a little modification for late production phase.

L102-103: 6 eggs were randomly chosen from each treatment in the experiment, and their external and internal quality were assessed. This is an extremely low number of eggs.

Thank you for highlighting that; I realize I didn't fully explain it in the manuscript. We used 3 eggs per replicate to assess quality parameters, pooling the data into a single value for analysis. In total, 18 samples from each group were evaluated.

Table 2 and 3: all significant 2 ways interactions need to be discussed in the result section.

We thank the reviewer for this comment.

From table 3: it seems that control was the best treatment, the additives lowered the performance.

Thank you for your question. Interestingly, B. coagulans and WP reduced performance when used alone compared to the control, but improved performance when mixed.

What is the difference between data in table5 and table 4?

All the tables have been revised accordingly. Before updating my article, I used the slice method to compare means after establishing the significance of the interaction effect, Considering the important points you taught me. The table in the article below has been used as a template for revising the revised tables.

L225-230: revise this part to reflect the results you got from the experiment.

Response: Thank you for your time and consideration. It was revised.

---

## [Decision Letter · Decision Letter 1]

25 Mar 2025

Effects of dietary of Bacillus coagulans, whey powder, and their interaction on the performance of Lohmann LSL-lite laying hens in the late production phase

PONE-D-24-54255R1

Dear Dr. Torki,

We’re pleased to inform you that your manuscript has been judged scientifically suitable for publication and will be formally accepted for publication once it meets all outstanding technical requirements.

Kind regards,

Shengqian Sun

Academic Editor

PLOS ONE

Additional Editor Comments (optional):

Reviewers' comments:

Reviewer's Responses to Questions

**Comments to the Author**

1. If the authors have adequately addressed your comments raised in a previous round of review and you feel that this manuscript is now acceptable for publication, you may indicate that here to bypass the “Comments to the Author” section, enter your conflict of interest statement in the “Confidential to Editor” section, and submit your "Accept" recommendation.

Reviewer #1: All comments have been addressed

Reviewer #2: All comments have been addressed

Reviewer #3: All comments have been addressed

2. Is the manuscript technically sound, and do the data support the conclusions?

Reviewer #1: Yes

Reviewer #2: Yes

Reviewer #3: Yes

3. Has the statistical analysis been performed appropriately and rigorously? 

Reviewer #1: Yes

Reviewer #2: Yes

Reviewer #3: Yes

4. Have the authors made all data underlying the findings in their manuscript fully available?

Reviewer #1: Yes

Reviewer #2: Yes

Reviewer #3: Yes

5. Is the manuscript presented in an intelligible fashion and written in standard English?

Reviewer #1: Yes

Reviewer #2: Yes

Reviewer #3: Yes

6. Review Comments to the Author

Reviewer #1: No further observations

The authors have carefully revised the manuscript. It can be accepted for publication.

Reviewer #2: (No Response)

Reviewer #3: Effects of dietary of Bacillus coagulans, whey powder, and their interaction on the performance of Lohmann LSL-lite laying hens in the late production

phase Thank you for providing a revised version of the manuscript. No further comments

7. PLOS authors have the option to publish the peer review history of their article (what does this mean? ). If published, this will include your full peer review and any attached files.

**Do you want your identity to be public for this peer review?** For information about this choice, including consent withdrawal, please see our Privacy Policy .

Reviewer #1: No

Reviewer #2: **Yes: ** Hassan Saleh

Reviewer #3: **Yes: ** Ala Abudabos

---

## [Editor Report · Acceptance letter]

PONE-D-24-54255R1

PLOS ONE

Dear Dr. Torki,

I'm pleased to inform you that your manuscript has been deemed suitable for publication in PLOS ONE. Congratulations! Your manuscript is now being handed over to our production team.

Kind regards,

on behalf of

Dr. Shengqian Sun

Academic Editor

PLOS ONE